# Antimutagenicity and Antioxidant Activity of *Castanea sativa* Mill. Bark Extract

**DOI:** 10.3390/pharmaceutics15102465

**Published:** 2023-10-14

**Authors:** Sofia Gasperini, Giulia Greco, Sabrina Angelini, Patrizia Hrelia, Carmela Fimognari, Monia Lenzi

**Affiliations:** 1Department of Pharmacy and Biotechnology, Alma Mater Studiorum Università di Bologna, Via San Donato 15, 40127 Bologna, Italy; sofia.gasperini4@unibo.it (S.G.); s.angelini@unibo.it (S.A.); patrizia.hrelia@unibo.it (P.H.); m.lenzi@unibo.it (M.L.); 2Department of Chemistry “Giacomo Ciamician”, Alma Mater Studiorum Università di Bologna, Via Selmi 2, 40126 Bologna, Italy; giulia.greco9@unibo.it; 3Department for Life Quality Studies, Alma Mater Studiorum Università di Bologna, Corso d’Augusto 237, 47921 Rimini, Italy

**Keywords:** *Castanea sativa* Mill., cancer chemoprevention, antimutagenesis, micronucleus, antioxidant, ROS, TK6, flow cytometry

## Abstract

*Castanea sativa* Mill. (Cs), a plant traditionally employed in nutrition and to treat various respiratory and gastrointestinal infections, possesses cancer chemopreventive characteristics. In particular, Cs bark extract previously demonstrated antiproliferative and pro-apoptotic activities against a leukemic lymphoblastic cell line. Starting from this evidence, the aim of this paper was to investigate the possibility to affect also the earlier phases of the carcinogenic process by evaluating Cs bark extract’s antimutagenic properties, in particular using the “In Vitro Mammalian Cell Micronucleus Test” on TK6 cells performed by flow cytometry. For this purpose, since an ideal chemopreventive agent should be virtually nontoxic, the first step was to exclude the extract’s genotoxicity. Afterwards, the antimutagenic effect of the extract was evaluated against two known mutagens, the clastogen mitomycin C (MMC) and the aneugen vinblastine (VINB). Our results indicate that Cs bark extract protected cells from MMC-induced damage (micronuclei frequency fold increase reduction from 2.9 to 1.8) but not from VINB. Moreover, we demonstrated that Cs bark extract was a strong antioxidant and significantly reduced MMC-induced ROS levels by over 2 fold. Overall, our research supports the assumption that Cs bark extract can counteract MMC mutagenicity by possibly scavenging ROS production.

## 1. Introduction

Cancer chemoprevention centers on the use of agents that specifically affect the different stages of cellular transformation with the goal of preventing, interrupting or reversing the carcinogenic process [1,2].

Among cancer chemopreventive agents, antimutagens are compounds able to decrease or even remove induced or spontaneous mutations. This process, called antimutagenesis, can be crucial for many diseases in addition to cancer. In fact, mutagenic changes that occur in somatic cells may contribute to the pathogenesis of other pathological conditions as well, e.g., neurodegenerative [3] and cardiovascular [4] diseases, while mutations in germline cells can be transmitted to future generations, possibly causing heritable genetic disorders [5].

Antimutagenic compounds can act through various molecular modes of action. They can prevent the mutagen from reaching the target by inhibiting their penetration into the cell or cell nucleus or by reducing their absorption and distribution in the body or by accelerating their evacuation from the body. They can also inactivate mutagens by direct chemical interaction and scavenging activity [6,7]. Moreover, some compounds can inhibit the enzymatic activation of pro-mutagens into effective mutagens or increase the activity of detoxifying enzymes [7]. Another antimutagenic mechanism is based on the enhancement of DNA repair mechanisms that can be augmented and accelerated in healthy cells, helping to maintain genomic stability [8].

In addition, as various mutagens also act through the generation of reactive oxygen species (ROS), which may react with DNA possibly causing a mutation [5,9], a strategy to reduce genetic damage is related to antioxidant processes, e.g., the inhibition of free radical oxidation, the scavenging of ROS, the chelation of transition metals and the increase in enzymatic antioxidant defense [7].

Finally, less studied antimutagenic mechanisms include the ability to increase the content of endogenous antimutagens (e.g., glutathione, melatonin, ubiquinones, interferons and bile acids) and neuroendocrine homeostasis regulation [7].

Antimutagenic agents often act through multiple mechanisms providing protection against diverse mutagens and, therefore, significantly improving the antimutagenic effectiveness. Thus, the search for such pleiotropic antimutagens is of great interest [5].

In this context, extracts from different parts of the plant *Castanea sativa* Mill. (*Cs*), a plant traditionally employed in nutrition and to treat conditions such as diarrhea and different respiratory diseases [10], have been shown to possess cancer chemopreventive characteristics. Of note, *Cs* possesses a strong antioxidant capacity, demonstrated for different parts of the plant in in vitro and in vivo models [10,11,12,13,14].

Other more specific cancer chemopreventive properties of *Cs* regard the ability of its bark extract to trigger the extrinsic apoptotic pathway in an acute leukemia cell line without appreciable cytotoxic effects on human peripheral blood lymphocytes from healthy donors, therefore suggesting a good selectivity of action for cancer cells [15]. Moreover, *Cs* leaf extract presented a protective effect against UV-mediated chromosomal damage in a human keratinocyte cell line probably due to a direct antioxidant effect [16]. Decoctions and infusions of *Cs* flowers were evaluated in four human tumor cell lines (breast adenocarcinoma MCF7, colon carcinoma HCT15, cervical carcinoma HeLa and hepatocellular carcinoma HepG2), highlighting an antiproliferative activity particularly against colon and hepatocellular cancer cell lines possibly related to two polyphenols: trigalloyl-HHDP-glucoside and pentagalloyl-glucoside [17]. As concerns in vivo studies, *Cs* flower extract showed antioxidant and anti-inflammatory properties in a model of prostate cancer [18] and the wood extract efficiently protected blood lymphocytes against DNA oxidative damage in vivo [11].

Most *Cs* properties have been ascribed to the high polyphenols content of the various parts of *Cs*. In fact, *Cs* is a typical source of polyphenols, mainly tannins, in the form of hydrolyzable tannins (ellagitannins, gallotannins), found in bark, burs, flours and leaves, and condensed tannins (procyanidins), the only form contained in chestnut peel. In particular, the main ellagitannins found in the bark are vescalagin and castalagin [19,20]. These components and antioxidants, contained in leaves, shells and wood, are already employed in animal feeds and may also be exploited in functional foods for humans, supporting recycling and valorizing vegetable waste in the perspective of a more environmentally friendly food production [19].

In this research, we focused on Cs bark extract’s cancer chemopreventive properties, specifically addressing its antimutagenic potential. For this purpose, since an ideal chemopreventive agent should be virtually nontoxic, it was essential to exclude the extract’s genotoxicity first. Afterwards, we tested its genoprotective effects against two mutagens, i.e., mitomycin C (MMC, a known clastogen agent) and vinblastine (VINB, a known aneugen agent). Moreover, given the known antioxidant properties of several molecules present in this extract, the cytoprotective effect against ROS has also been tested as a possible antimutagenesis mechanism.

## 2. Materials and Methods

### 2.1. Reagents

BPC-grade water, 2,7-dichlorodihydrofluorescin diacetate (DCFH-DA), dimethyl sulfoxide (DMSO), fetal bovine serum (FBS), hydrogen peroxide (H_2_O_2_), L-glutamine (L-GLU), mitomycin C (MMC), penicillin-streptomycin solution (PS), phosphate-buffered saline (PBS), Roswell Park Memorial Institute (RPMI) 1640 medium, sodium chloride, sodium hydrogen phosphate, vinblastine (VINB) (all purchased from Merck, Darmstadt, Germany), Guava Nexin reagent, Guava ViaCount reagent (purchased from Luminex Corporation, Austin, TX, USA), and SYTOX Green (purchased from Thermo Fisher Scientific, Waltham, MA, USA) were used.

### 2.2. Castanea sativa Mill. Bark Extract

Cs bark extract was obtained by SilvaTeam (San Michele di Mondovì, Italy) by a low-pressure heating treatment and supplied as dry extract in the form of a dark brown powder [21]. The extract was previously characterized by Chiarini et al. by HPLC-DAD-MS analysis that revealed high levels of phenolic compounds such as castalin, vescalin, castalagin, vescalagin, ellagic acid and gallic acid as reported in their publication [12].

The powder was preserved at room temperature protected from light. Prior to each experiment, the powder was solubilized at a concentration of 10 mg/mL in a solvent mix composed of 80% *v*/*v* RPMI 1640 and 20% *v*/*v* DMSO. To avoid DMSO toxicity, its final concentration in cell cultures never exceeded 1% *v*/*v*.

### 2.3. Cell Culture

All endpoints were evaluated on human lymphoblastoid TK6 cells, a cell line selected among those validated by the Organization for Economic Co-operation and Development (OECD) to perform micronucleus (MN) test, by virtue of its human and nontumoral origin and replicative speed [22,23].

TK6 cells (ATCC, Manassas, VA, USA) were cultured at 37 °C and 5% CO_2_ in a mix of: 90% RPMI-1640, 10% FBS, 1% L-GLU and 1% PS. TK6 cells require 13 h to duplicate. Cell density should not exceed the critical value of 9 × 10^5^ cells/mL. In each experiment, aliquots of 2.5 × 10^5^ TK6 cells were treated.

### 2.4. Selection of Concentrations for MNi Frequency Measurement

Cytotoxicity and the cytostasis threshold reported in OECD guideline No. 487 are equal to 55 ± 5% [22]. Consequently, the concentrations for the MN test should determine a viability and a proliferation of at least 45 ± 5%. The treatment duration (26 h) corresponds to the time required by TK6 cells to complete 1.5–2.0 normal cell cycles cells, which is necessary to transmit the genetic damage possibly suffered to daughter cells, with different Cs bark extract concentrations as specified below.

#### 2.4.1. Measurement of Cytotoxicity

Cytotoxicity was measured in samples treated with the following Cs bark extract concentrations: 0, 3, 6, 12, 18, 24, 48 μg/mL. Cytotoxicity was measured using the Guava ViaCount Reagent and performing Guava ViaCount Assay, according to the manufacturer’s instructions and previous publications [24,25,26]. In total, 1000 events were acquired. The viability percentage and the number of total live cells in the original sample were then automatically calculated by the Guava ViaCount software 2.7 (Merck, Darmstadt, Germany). The viability percentage calculated was normalized to that of the untreated control, considered 100%.

#### 2.4.2. Measurement of Cytostasis

Cytostasis was measured in samples treated with the following Cs bark extract concentrations: 0, 3, 6, 12, 18, 24, 48 μg/mL. Cytostatsis was checked as follows. From the number of cells seeded and that detected by means of the Guava ViaCount assay at the end of the treatment (performed according to the manufacturer’s instructions and previous publications [24,27,28]), cytostasis was measured and cell replication was calculated as Population Doubling (PD):$$PD=log\left(\frac{post-treatment\;cell\;number}{initial\;cell\;number}\right)÷log2$$

Afterward, the PD of the untreated controls and that of treated cultures were used to calculate Relative Population Doubling (RPD) and to check that at least 45 ± 5% of cells had replicated after treatment time.
$$RPD=\frac{PD\;in\;treated\;cultures}{PD\;in\;control\;cultures}\times 100$$

#### 2.4.3. Measurement of Apoptosis

Apoptosis was measured in samples treated with the following Cs bark extract concentrations: 0, 3, 6, and 12 μg/mL, to more accurately select Cs bark extract concentrations to be used in the MN test, so that they do not induce more than a doubling in the apoptosis levels compared to those recorded in the concurrent negative cultures.

As such, aliquots of 2.5 × 10^5^ TK6 cells were treated with Cs bark extract concentrations selected based on cytotoxicity and cytostasis results (0, 3, 6, and 12 μg/mL) for 26 h.

Apoptosis was measured was using the Guava Nexin Reagent and performing the Guava Nexin Assay according to the manufacturer’s instructions and previous publications [29,30,31]. For this assay, 2000 events were acquired and

The percentage of apoptosis cells was normalized to that of the untreated control, considered 1 and reported as the apoptotic fold increase.

### 2.5. Measurement of Mni Frequency

Mni frequency was measured in samples treated with the following Cs bark extract concentrations: 0, 3, 6, and 12 μg/mL, chosen starting from the results of abovementioned analyses. MMC and VINB were employed as positive controls [22].

After the treatment, the Mni frequency was assessed by a flow cytometry (FCM) according to the protocol published by Lenzi et al. [32] and already used in previous publications [22,25,31]. The Mni frequency was normalized to that of the untreated control, considered 1 and reported as the Mni frequency fold increase.

For the analysis of the possible antimutagenic potential of Cs bark extract, TK6 cells were co-treated with the selected Cs bark extract concentrations (0, 3, 6, and 12 μg/mL) and MMC or VINB. After the treatment, cytotoxicity, cytostasis and apoptosis levels were checked according to the thresholds previously described and afterwards the same MN assay protocol was performed.

### 2.6. Measurement of Intracellular ROS Levels

TK6 cells were treated, both in the absence and presence of an exogenous oxidative stress such as H_2_O_2_, for a short treatment time of 6 h since ROS production is an extremely early cellular event, with the same Cs bark extract concentrations used in the genotoxicity test (0, 3, 6, and 12 μg/mL) and, in addition, a higher concentration (24 μg/mL) in consideration of the much shorter treatment time (6 h vs. 26 h) [33].

Moreover, to investigate the possible protective effect of Cs bark extract against the ROS generated by the mutagen MMC, aliquots of 2.5 × 10^5^ TK6 cells were first treated with MMC alone for 6 h or 12 h to select the correct treatment time at which MMC induces ROS in TK6 cells. Afterwards, TK6 cells were treated with the same Cs bark extract concentrations used in the genotoxicity test (0, 3, 6, and 12 μg/mL), both in the absence and presence of MMC 200 ng/mL, for 12 h.

At the end of the treatment, once we checked that cell viability was as described in paragraph 2.4.1, intracellular ROS levels were measured by employing 2,7-dichlorodihydrofluorescein diacetate (DCFH-DA) and by the Guava InCyte Assay. More specifically, cells were resuspended in PBS and incubated for 20 min with DCFH-DA 10 μM and then half of the samples were also exposed to H_2_O_2_ 100 μM for an additional 20 min. For each sample, 5000 events were acquired. Dichlorofluorescin (DCF), a brightly green fluorescent molecule, is formed in cells in the presence of ROS [31,34,35]. The mean green fluorescence intensity measured in treated cultures was normalized on that measured in negative controls, set equal to 1, and expressed as the ROS fold increase.

The results were also reported in terms of the percentage of cells at low or high fluorescence intensity.

H_2_O_2_ 100 μM was also employed as a positive control.

### 2.7. Flow Cytometry

FCM analyses were carried out with a Guava EasyCyte 5HT flow cytometer—class IIIb laser operating at 488 nm (Luminex Corporation, Austin, TX, USA).

### 2.8. Statistical Analysis

Three independent experiments were performed for all analyses and results are reported as the mean ± standard error of the mean (SEM). ANOVA repeated (for ROS analyses) or mixed-effects analysis (for all other analyses), followed by the Dunnett or Bonferroni post-test, was used to evaluate the statistical significance (Prism Software 9.0). According to GraphPad Prism, manual results were considered statistically significant if *p* < 0.05.

## 3. Results

### 3.1. Cytotoxicity

In the first phases of the research, cytotoxicity and cytostasis tests have been performed to select the most suitable Cs bark extract concentrations for the subsequent Mni frequency evaluation. Cell viability must be greater than 45 ± 5%, i.e., the threshold set by the OECD, if compared to that of control cultures, set as 100% [22].

Figure 1 displays TK6 viability after Cs bark extract treatments. Viability percentages were above the OECD threshold (represented by the red line), up to a concentration of 18 µg/mL.

### 3.2. Cytostasis

Since a sufficient number of cells should undergo mitotic activity to be able to transmit the genetic damage possibly suffered by daughter cells, we measured cell replication as RPD. RPD must be greater than 45 ± 5%, i.e., the threshold set by the OECD, if compared to that of control cultures, set as 100% [22]. Only concentrations up to 12 µg/mL complied with the OECD threshold (Table 1).

### 3.3. Apoptosis

Based on cytotoxicity and cytostasis results, 0, 3, 6, and 12 µg/mL Cs bark extract concentrations have also been tested for the apoptosis analysis. Indeed, apoptotic bodies could be “mistaken” and erroneously counted as MNi by the instrument [32]. Figure 2 shows that apoptosis levels were comparable to those of the negative control at any of the tested concentrations.

### 3.4. Measurement of MNi Frequency

Thereby, based on the results obtained for cytotoxicity, cytostasis and apoptosis, the Cs bark extract concentrations selected for the MNi frequency evaluation were 3, 6 and 12 µg/mL.

First, to rule out the potential mutagenic effect associated with the extract, the MNi frequency was measured in untreated cultures (negative controls), in cultures treated with Cs bark extract and in cultures treated with the known mutagens MMC and VINB (positive controls). As reported in Figure 3, the increase in the MNi frequency recorded in the cultures treated with Cs bark extract was not statistically significant compared to the negative control, while a 2 or 3-fold increase was detected in the cultures treated with MMC or VINB (Figure 3A). This outcome can also be observed in the representative dot plots for the negative control and Cs bark extract 12 µg/mL, highlighting a similar low numerosity of MNi for both conditions (Figure 3B,C).

Once the nonmutagenicity of the extract had been demonstrated, this study continued by evaluating its possible antimutagenic activity against the two known mutagens previously used as positive controls (MMC and VINB).

For this purpose, a 26 h co-treatment was performed and, also in this case, before proceeding with the genotoxicity analysis, cytotoxicity, cytostasis, and apoptosis were analyzed in order to check the viability and the correct replication in the cell cultures.

As shown in Figure 4 and Table 2, respectively, cell viability and RPD were well above the threshold established by the OECD (represented by the red line).

With regard to apoptosis, its levels after the co-treatments were still comparable to those of the untreated negative control in all treatment conditions (Figure 5).

Overall, the results obtained allowed us to proceed with the MNi frequency evaluation testing the three Cs bark extract concentrations selected (3, 6, and 12 µg/mL) in association with the clastogen MMC or the aneugen VINB.

In Figure 6A, a decreasing trend in cultures co-treated with MMC and the three concentrations of Cs bark extract when compared to cultures treated only with MMC could be noted. The result was statistically significant at the Cs bark extract concentrations 6 and 12 µg/mL (MNi frequency fold increase reduction from 2.9 in MMC to 1.8 in Cs bark extract 6 + MMC and Cs bark extract 12 + MMC).

As regards VINB co-treatments, the outcomes showed that Cs bark extract at the concentrations 3 and 6 µg/mL was not able to decrease the MNi frequency induced by VINB, while a slight decrease compared to cultures treated only with VINB was observed at the concentration 12 µg/mL. However, this tendency was not observed in all replicates and, therefore, it is not significant (Figure 6D).

These results can also be observed in the representative dot plots for MMC 200 ng/mL and Cs bark extract 6 µg/mL + MMC 200 ng/mL (Figure 6B,C) and VINB 6.25 ng/mL and Cs bark extract 6 µg/mL + VINB 6.25 ng/mL (Figure 6E,F).

### 3.5. Measurement of Intracellular ROS Levels

This test has been performed to investigate a possible antimutagenic mechanism, i.e., the ability to scavenge intracellular ROS. In fact, many mutagens, including MMC, also owe their effect to the generation of intracellular ROS that can damage the genetic material.

To investigate whether Cs bark extract could have a protective antioxidant effect, possibly linked to the proven antimutagenic capacity against MMC, concentrations of Cs bark extract (0, 3, 6, 12, 24 µg/mL) were tested after a short treatment time of 6 h, since ROS production is an extremely early cellular event. H_2_O_2_ 100 µM was used as a positive control.

At the end of the treatment time, after checking cellular viability (Figure 7), ROS intracellular levels were evaluated as the mean green fluorescence intensity of DCF.

The mean green fluorescence measured in treated cultures was then compared to that of the negative controls, set equal to 1.

The results highlighted antioxidant activity as Cs bark extract diminished ROS intracellular basal levels in a concentration-dependent manner, achieving a statistically significant decrease in the mean green fluorescence intensity at the Cs bark extract concentration 24 µg/mL (Figure 8).

This outcome suggested a possible ability of Cs bark extract to also counteract exogenous oxidative stress. To examine this hypothesis, cells were pretreated with Cs bark extract for 6 h and then exposed to H_2_O_2_ 100 µM. Cs bark extract treatment reduced ROS production already at the concentration 12 µg/mL, while it completely neutralized the production of ROS by H_2_O_2_ at 24 µg/mL, leading to a statistically significant decrease compared to H_2_O_2_ 100 µM treatment (Figure 8).

Considering the antioxidant effect demonstrated, we hypothesized that one possible mechanism by which Cs bark extract hinders MMC mutagenic action could be to scavenge the ROS possibly produced by this mutagen.

For this purpose, first, we treated TK6 cells for 6 or 12 h with MMC to check whether MMC actually induced ROS in our experimental settings.

Figure 9 shows that MMC 200 ng/mL was able to increase ROS levels after 12 h, as demonstrated by the shift in the fluorescence peak in the histogram.

Afterwards, we co-treated TK6 cells with both MMC and Cs bark extract at the same concentrations used in the genotoxicity test. The results showed that Cs bark extract had an antioxidant effect, particularly at the concentration of 12 µg/mL (reported in Figure 10). Indeed, we recorded a 5-fold increase in the cell percentage at low fluorescence intensities and a simultaneous over 2-fold decrease in the cell percentage at high fluorescence intensities in samples co-treated with Cs bark extract plus MMC compared to the samples treated only with MMC (Figure 10A,B). These results suggest that Cs bark extract was able to counteract MMC-induced ROS production.

## 4. Discussion

The aim of this paper was to investigate the ability of Cs bark extract to exert antimutagenic activity, thus inhibiting the earlier phases of the carcinogenic process. The first step of our research was to verify its safety of use, also in terms of mutagenicity.

According to our knowledge, very little information is available concerning Cs bark extract genotoxicity. In fact, bibliographic research conducted on the main databases (i.e., PubMed from MEDLINE and Scopus from Elsevier) allowed us to identify only two publications on this specific subject. In particular, Brus and collaborators investigated the potential genotoxicity of a water-soluble Cs bark extract, a tannin extract, by labeling chicken small intestinal epithelial cells with an antibody against histone H2AX, which reveals DNA breakages and exposed histones within the nucleus. They found no statistically significant differences between controls and extract-treated cells [36]. Almeida et al. observed no genotoxic effect of Cs bark extract and leaf extract in a human keratinocyte cell line (HaCaT) using the cytokinesis-block MN assay [16]. Moreover, as concerns isolated compounds that could be present in Cs bark extract extracts, it was also shown that tannic, ellagic, and gallic acids were not mutagenic in the Ames test [37,38], while they were found to be genotoxic, but only at high concentrations, in the comet assay performed on trout erythrocytes and digestive cells of freshwater mussels [39,40]. In another study, tannic and gallic acids caused oxidative strand breakage in DNA either alone or in the presence of transition metal ions such as copper [41,42]. Lastly, there is some evidence regarding castalagin and vescalagin inhibitory activity against PARP1, a protein also involved in DNA repair and transcriptional regulation, and Topoisomerase II, a protein that manages DNA tangles and supercoils by cutting and rejoining double-stranded DNA [43,44].

In light of these diverse data, we tested the mutagenic activity of Cs bark extract before proceeding with the analysis of its antimutagenic potential.

Hence, we started our work evaluating the cytotoxic activity (in terms of a reduction in cell viability and proliferation and induction of apoptosis) of Cs bark extract. This step was necessary to define the noncytotoxic concentrations to be used for the subsequent evaluation of its mutagenic activity, which was performed through the MN test on TK6 cells. MN is recognized as an important biomarker of chromosomal damage, genomic instability and cancer risk. TK6 cells were selected for their human and nontumor origin among those recommended by OECD guideline No. 487, corresponding to the “in vitro mammalian cell micronucleus test” [22].

The mutagenic action was then evaluated and calculated as the increase in the MNi frequency measured by FCM, an alternative platform, which allows overcoming the typical limitations of the classical scoring method by optical microscopy, such as the limited number of cells analyzed, the long analysis times, and the high subjectivity in the interpretation of the results by the operator [32].

After demonstrating the absence of the mutagenicity of Cs bark extract, we investigated its antimutagenic potential. Currently, a few studies are available on the antigenotoxic effects of Cs bark extract obtained from different plant parts. In particular, a protective effect of Cs bark extract and leaf extract was demonstrated against UV-mediated chromosomal damage, measured using the MN assay, in HaCaT cells and this property was ascribed to a direct antioxidant effect [16]. Further, in young pigs fed with high and pro-oxidant n-3 polyunsaturated fatty acid (PUFA) quantities, Cs bark extract, the wood extract, limited DNA oxidative damage in blood lymphocytes, analyzed with the alkaline comet assay [11].

Our study aimed at further investigating Cs bark extract antimutagenic potential in terms of the decrease in the frequency of MNi induced by two known mutagenic agents acting with different mechanisms: the clastogen MMC and the aneugen VINB.

Our results indicate that Cs bark extract protected cells from MMC- but not from VINB-induced damage. In fact, a greater protective effect was observed when Cs bark extract was associated with MMC rather than with VINB. Even if our findings are surely preliminary, as only two mutagens were included in this study, they nevertheless allow highlighting the different behaviors of Cs bark extract: Cs bark extract might be able to counteract structural DNA damage rather that genomic DNA damage.

This diverse outcome encouraged us to hypothesize a possible explanation for Cs bark extract activity based on the complex mode of action characterizing MMC toxicity. Indeed, this clastogen can generate monoalkylation or dialkylation products and form covalent cross-linking between DNA complementary strands. This interaction prevents strand separation, inhibits DNA replication, and causes its break [45]. Furthermore, MMC generates ROS such as ^1^O_2_, H_2_O_2_, and OH*. Thus, the association of MMC with antioxidant molecules represents a possible approach to prevent MMC-induced DNA damage [46,47]. Since the antioxidant properties of Cs bark extract have long been demonstrated [10,11,12,13,14], it is plausible to hypothesize that it was able to lower ROS levels and counteract MMC genotoxicity.

We tested this hypothesis by exploring the antioxidant capacity of the extract (which was never performed before) and then measuring ROS levels both after MMC treatment and Cs bark extract–MMC co-treatment. MMC increased ROS levels in the TK6 cell line and Cs bark extract significantly reduced MMC-induced ROS levels, thus supporting the assumption that Cs bark extract could counteract MMC mutagenicity by possibly scavenging ROS production. This outcome also agrees with the results of Almeida et al., who reported the ability of Cs bark extract and leaf extract to counteract UV-mediated MNi induction in HaCaT cells probably due to a direct antioxidant effect rather than the activation of endogenous antioxidant responses [16].

Although preliminary, our results support an interesting biological potential for Cs bark extract. This should be confirmed using different clastogens to verify if it is common against all clastogens and/or ROS-producing agents or if it is a peculiar behavior against MMC-induced DNA damage.

Moreover, since our experiments were carried out in a co-treatment condition, our results could also suggest a direct extracellular interaction between the extract and MMC. Further research will be performed pretreating cells with Cs bark extract and after with MMC to verify if the genoprotection is also potentiated through an indirect antioxidant effect.

## 5. Conclusions

Overall, our research supports the assumption that Cs bark extract can protect cells from MMC- but not from VINB-induced chromosomal damage, evaluated as a reduction in MNi frequency, by possibly scavenging MMC-induced ROS production.

## Figures and Tables

**Figure 1 pharmaceutics-15-02465-f001:**
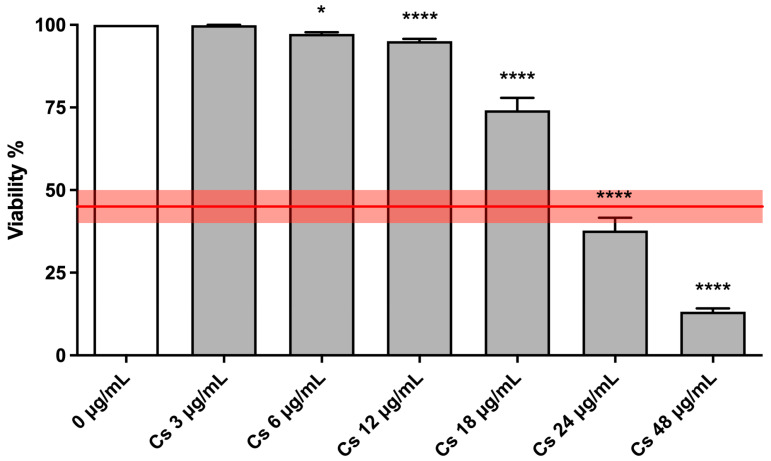
Cell viability of TK6 cells after a 26 h treatment at the indicated concentrations compared to the concurrent negative control (0 µg/mL). The red line represents the OECD threshold for viability (45 ± 5%). * *p* < 0.05 vs. (0 µg/mL); **** *p* < 0.0001 vs. (0 µg/mL).

**Figure 2 pharmaceutics-15-02465-f002:**
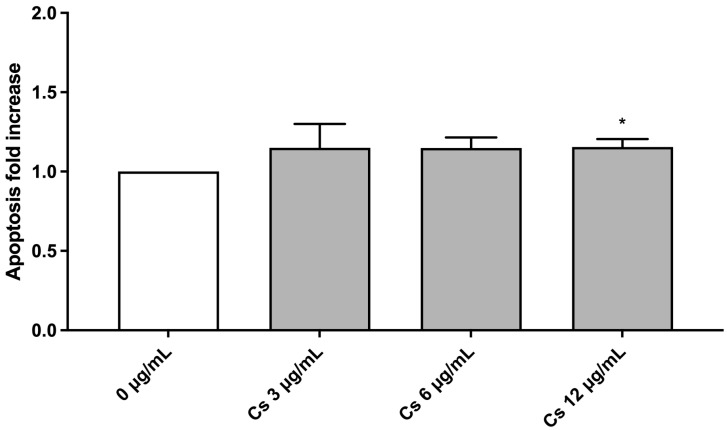
Apoptosis fold increase on TK6 cells after a 26 h treatment at the indicated concentrations compared to the concurrent negative control (0 µg/mL). * *p* < 0.05 vs. (0 µg/mL).

**Figure 3 pharmaceutics-15-02465-f003:**
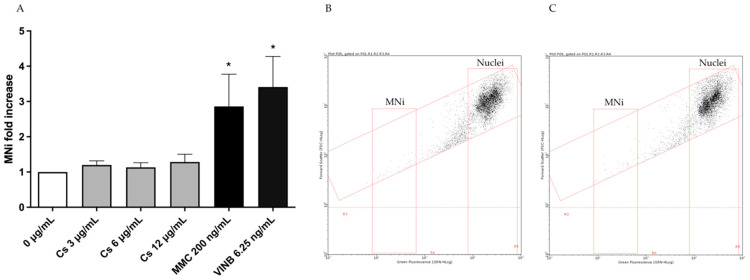
(**A**) MNi fold increase on TK6 cells after a 26 h treatment at the indicated concentrations compared to the concurrent negative control (0 µg/mL). * *p* < 0.05 vs. (0 µg/mL). Representative FCM dot plot of nuclei and MNi (**C**) in cultures treated with Cs bark extract 12 µg/mL and (**B**) in the concurrent negative control culture.

**Figure 4 pharmaceutics-15-02465-f004:**
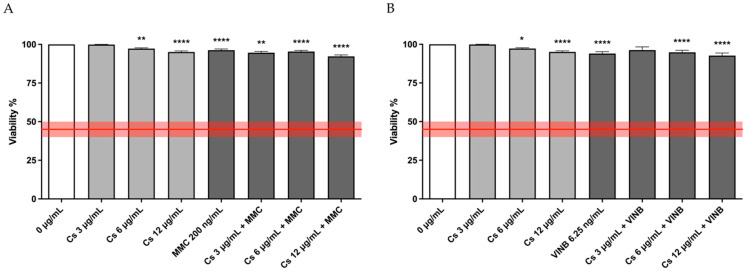
Cell viability of TK6 cells after a 26 h treatment with (**A**) MMC or (**B**) VINB plus Cs bark extract at the indicated concentrations compared to the concurrent negative control (0 µg/mL). The red line represents the OECD threshold for viability (45 ± 5%). * *p* < 0.05 vs. (0 µg/mL); ** *p* < 0.01 vs. (0 µg/mL); **** *p* < 0.0001 vs. (0 µg/mL).

**Figure 5 pharmaceutics-15-02465-f005:**
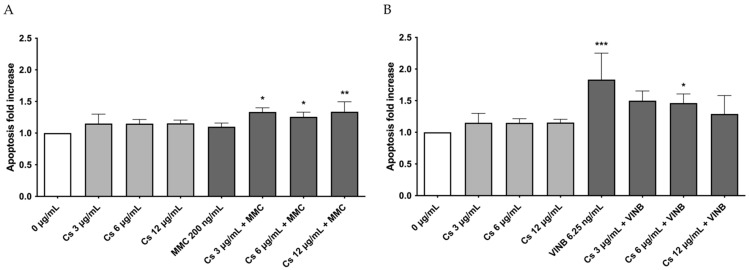
Apoptosis fold increase in TK6 cells after a 26 h treatment with (**A**) MMC or (**B**) VINB plus Cs bark extract at the indicated concentrations compared to the concurrent negative control (0 µg/mL). * *p* < 0.05 vs. (0 µg/mL); ** *p* < 0.01 vs. (0 µg/mL); *** *p* < 0.001 vs. (0 µg/mL).

**Figure 6 pharmaceutics-15-02465-f006:**
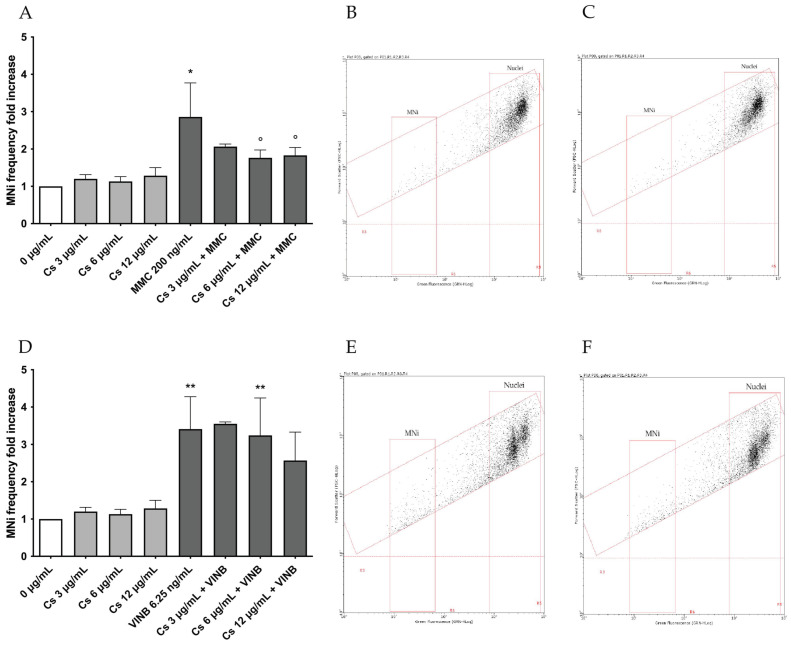
MNi fold increase on TK6 cells after a 26 h treatment at the indicated concentrations compared to the concurrent negative control (0 µg/mL) and to the positive control MMC (**A**) or VINB (**D**). * *p* < 0.05 vs. (0 µg/mL); ** *p* < 0.01 vs. (0 µg/mL); ° *p* < 0.05 vs. (MMC 200 ng/mL). Representative FCM dot plot of nuclei and MNi in cultures treated with MMC 200 ng/mL (**B**), in cultures treated with Cs bark extract 6 µg/mL plus MMC 200 ng/mL (**C**), VINB 6.25 ng/mL (**E**), and Cs bark extract 6 µg/mL plus VINB 6.25 ng/mL (**F**).

**Figure 7 pharmaceutics-15-02465-f007:**
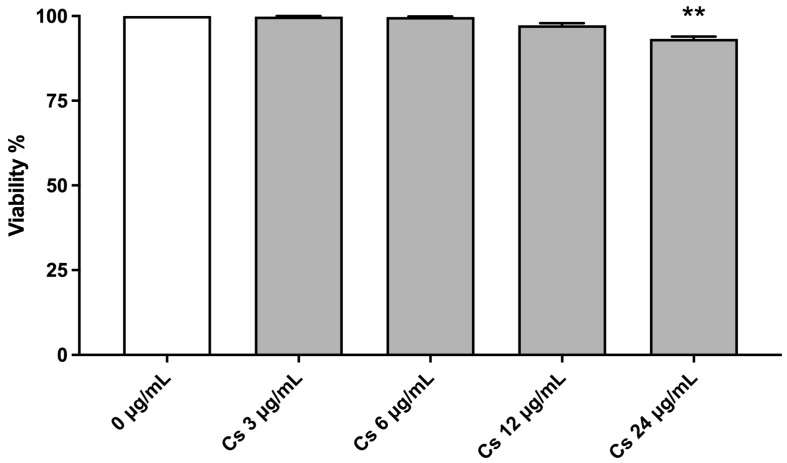
Cell viability on TK6 cells after 6 h treatment at the indicated concentrations compared to the concurrent negative control (0 µg/mL). ** *p* < 0.01 vs. (0 µg/mL).

**Figure 8 pharmaceutics-15-02465-f008:**
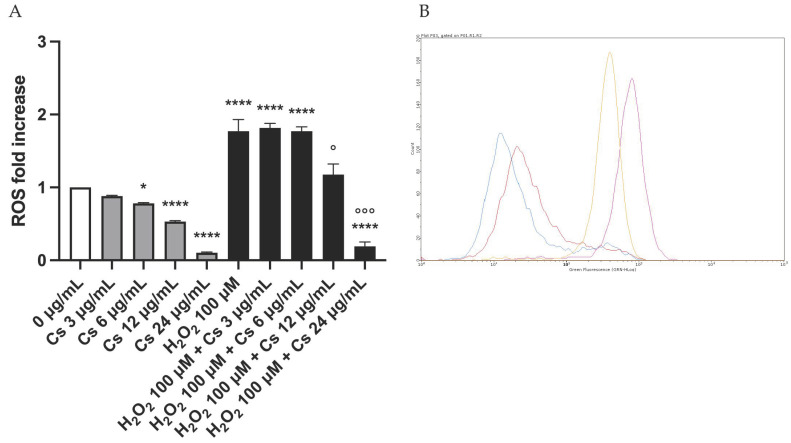
ROS fold increase on TK6 cells after 6 h treatment with Cs bark extract and/or the positive control (H_2_O_2_) at the indicated concentrations compared to the concurrent negative control (0 µg/mL). * *p* < 0.05 vs. (0 µg/mL); **** *p* < 0.0001 vs. (0 µg/mL). ° *p* < 0.05 vs. (H_2_O_2_ 100 μM) °°° *p* < 0.001 vs. (H_2_O_2_ 100 μM) (**A**). Representative FCM histogram of 0 µg/mL (orange), Cs bark extract 24 μg/mL (blue), H_2_O_2_ 100 μM (pink) and Cs bark extract 24 µg/mL + H_2_O_2_ (red) (**B**).

**Figure 9 pharmaceutics-15-02465-f009:**
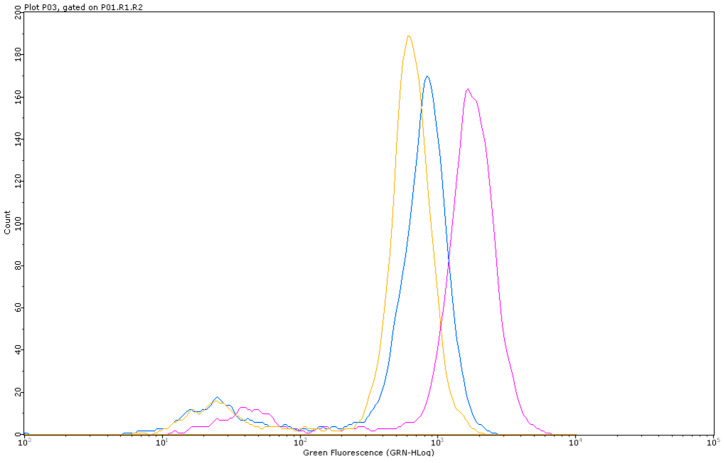
Representative FCM histogram of the green fluorescence intensity of 0 µg/mL (orange), MMC 200 ng/mL (blue) and H_2_O_2_ 100 μM (pink) on TK6 cells after a 12 h treatment.

**Figure 10 pharmaceutics-15-02465-f010:**
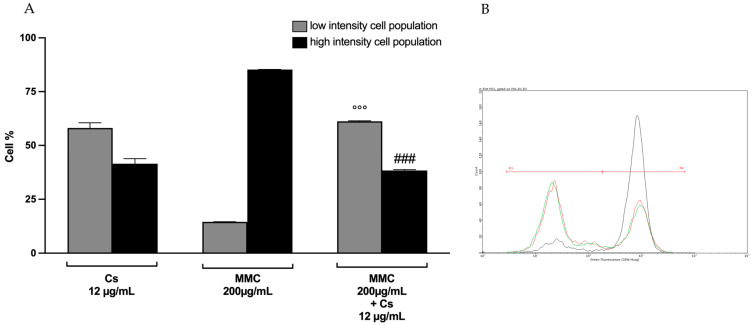
Percentage of cells at low or high fluorescence intensities after a 12 h treatment on TK6 cells with MMC plus Cs bark extract at the indicated concentrations. °°° *p* < 0.001 vs. (MMC 200 ng/mL, low intensity); ### *p* < 0.001 vs. (MMC 200 ng/mL, high intensity) (**A**). Representative FCM histogram of the green fluorescence intensity of cultures treated with Cs bark extract 12 μg/mL (red), MMC 200 ng/mL (black), and MMC 200 ng/mL plus Cs bark extract 12 μg/mL (green). R3 and R4 gates individuate the low and high fluorescence cell populations, respectively (**B**).

**Table 1 pharmaceutics-15-02465-t001:** RPD of TK6 cells after a 26 h treatment at the indicated concentrations compared to the concurrent negative control (0 µg/mL). **** *p* < 0.0001 vs. (0 µg/mL).

Cs Bark Extract Concentrations (µg/mL)	RPD
0	100.00%
3	97.95 ± 1.74%
6	94.20 ± 1.16%
12	81.47 ± 3.24% ****
18	19.66 ± 7.86% ****
24	0 ± % ****
48	0 ± % ****

**Table 2 pharmaceutics-15-02465-t002:** RPD of TK6 cells after a 26 h treatment with MMC or VINB plus Cs bark extract at the indicated concentrations compared to the concurrent negative control (0 µg/mL). At least three independent experiments have been performed and the collected results are reported as the mean ± SEM. Data were analyzed by mixed-effects analysis followed by the Dunnet post-test. ** *p* < 0.01 vs. (0 µg/mL); **** *p* < 0.0001 vs. (0 µg/mL).

Concentrations	RPD
0 µg/mL	100.00%
Cs bark extract 3 µg/mL	97.95 ± 1.74%
Cs bark extract 6 µg/mL	94.20 ± 1.15%
Cs bark extract 12 µg/mL	81.47 ± 3.24% ****
MMC 200 ng/mL	70.13 ± 2.78% ****
Cs bark extract 3 µg/mL + MMC 200 ng/mL	71.20 ± 1.51% ****
Cs bark extract 6 µg/mL + MMC 200 ng/mL	69.77 ± 2.90% ****
Cs bark extract 12 µg/mL + MMC 200 ng/mL	61.50 ± 2.55% ****
VINB 6.25 ng/mL	76.09 ± 5.55% **
Cs bark extract 3 µg/mL + VINB 6.25 ng/mL	78.67 ± 4.98% ****
Cs bark extract 6 µg/mL + VINB 6.25 ng/mL	76.80 ± 5.38% ****
Cs bark extract 12 µg/mL + VINB 6.25 ng/mL	70.52 ± 4.02% ****

## Data Availability

The data presented in this study are available within this article.

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
