# Peer review of "Antimutagenicity and Antioxidant Activity of *Castanea sativa* Mill. Bark Extract"

_pharmaceutics, 2023, doi:10.3390/pharmaceutics15102465_

Round 1
Reviewer 2 Report
The authors in the paper titled “Antimutagenic and antioxidant potential of Castanea sativa Mill. Bark extract‚‚ investigate the ability of CSM bark extract to exert an antimutagenic activity, thus inhibiting the earlier phases of the carcinogenic process. I recommend the paper for publishing after some minor revisions.

Reviewer 3 Report
The manuscript is written very solidly and contains valuable data. The authors could only mention the contents of the individual compounds in the extract (Table 1 in Reference 12). Is anything known about the antimutagenic activity of castalin, vescalin, castalagin, vescalagin? The authors only discuss the activity of ellagic and gallic acids.
The histograms should be of better quality, especially the axis descriptions.
Round 2
Reviewer 1 Report
Congratulations on your manuscript.